# Geodesic Flow Matching for Denoising High-Dimensional Structured Representations

**Karim Habashy** [1 2]  **Chris Eliasmith** [1 2 3]

## Abstract

Vector Symbolic Algebras (VSAs) enable robust neurosymbolic reasoning by encoding symbolic information into high-dimensional distributed representations. For continuous domains, Spatial Semantic Pointers (SSPs) extend this framework by mapping variables onto continuous toroidal manifolds. However, standard approaches like Flow Matching assume a flat Euclidean geometry, which fails to account for the geometric constraints imposed on valid SSP states. We demonstrate that this assumption fails for SSPs: Euclidean linear interpolants "cut through" the manifold's interior, destroying the phase and magnitude structure required for accurate decoding. To resolve this, we employ Geodesic Flow Matching, adapting Riemannian transport dynamics to strictly restrict the denoising flow to the SSP toroidal manifold. We validate this approach in a Spiking Neural SLAM system, showing that manifold-aware cleanup stabilizes path integration against drift. The method achieves a 72% reduction in tracking error and enables a 40% increase in neural efficiency compared to competitive baselines. Code is available at https://github.com/kremHabashy/CleanupSSP.

## 1. Introduction

Neurosymbolic AI aims to combine the robustness of neural networks with the structured compositionality of symbolic reasoning. Central to this approach are Vector Symbolic Algebras (VSAs), which encode symbolic information into high-dimensional distributed vectors. A critical requirement for robust operation of VSA-based systems is "cleanup": the ability to map noisy, composed, or partial inputs back to valid, clean states. While this is well-explored for discrete symbols (e.g., via Hopfield networks (Hopfield, 1982; Stewart et al., 2011; Ramsauer et al., 2020)), it remains an open challenge for continuous representations, where the "valid" states form a continuous manifold rather than a discrete set of attractors.

Recent theoretical work suggests that generative denoising serves as a modern, continuous form of pattern completion. Hoover et al. (2023) highlight the intersection between diffusion models and associative memories, where denoising score matching parallels energy minimization in attractor networks. Unlike classical models that suffer from limited capacity, generative transport models demonstrate emergent generalization, reconstructing valid states even in unexplored regions of the manifold (Pham et al., 2025).

However, translating these generative capabilities to real-time neurosymbolic systems presents distinct challenges. While diffusion models are powerful, they rely on iterative stochastic sampling, requiring many function evaluations that are prohibitively slow for low-latency tasks in robotics (e.g. Simultaneous Localization and Mapping (SLAM)). Conditional Flow Matching (CFM) (Lipman et al., 2022) solves this by regressing a deterministic velocity field (achieving comparable performance to diffusion with significantly fewer steps), but assumes a flat Euclidean geometry. Finally, recent work has successfully generalized this framework to Riemannian manifolds, enabling flow matching on general geometries (Chen & Lipman, 2024). However, these explorations have primarily focused on low-dimensional data and simple (gaussian) target distributions.

Recently, Spatial Semantic Pointers (SSPs) have been proposed as a continuous extension of VSAs that encodes spatial variables into high-dimensional vectors ($d > 1000$). Prior methods do not work to clean up SSPs because these representations reside on a highly structured Clifford Hypertorus embedded within the unit hypersphere $\mathbb{S}^{d-1}$. We demonstrate that standard Euclidean Flow Matching fails in this regime because its linear interpolants "cut through" the interior of the hypersphere. This leads to a collapse of

[1]Department of Systems Design Engineering, University of Waterloo, Waterloo, Canada [2]Center For Theoretical Neuroscience, University of Waterloo, Waterloo, Canada [3]Department of Philosophy, University of Waterloo, Waterloo, Canada. Correspondence to: Karim Habashy <khabashy@uwaterloo.ca>.

*Proceedings of the 43rd International Conference on Machine Learning*, Seoul, South Korea. PMLR 306, 2026. Copyright 2026 by the author(s).

vector magnitude and the destruction of the precise phase relationships required to restore the SSP encoding a particular location.

To resolve this, we employ Geodesic Flow Matching (GFM), adapting the framework to high-dimensional, toroidal neurosymbolic representations. By constraining transport to the Riemannian manifold using logarithmic and exponential maps, we ensure the cleanup process remains geometrically consistent. To demonstrate the methods in a challenging application, we have chosen a spiking neural network. Spiking networks provide useful efficiency gains when running on neuromorphic hardware, but introduce significant additional noise into neural representations that make robust behaviour challenging (Pfeiffer & Pfeil, 2018). Spiking networks are thus a difficult test for robust cleanup of neural representation. We achieve a 72% reduction in path error and a 40% increase in neural efficiency compared to competitive baselines. The application, Semantic SLAM (Dumont et al., 2023), serves as a continuous neurosymbolic benchmark: the full pipeline relies on VSA binding and unbinding operations whose correctness depends directly on the geometric fidelity of the cleaned representations.

Our work adapts flow matching to high-dimensional geodesics, providing a robust framework for hyperspherical embeddings (a geometry increasingly central to modern AI). This applies directly to architectures like Hyperspherical Prototype Networks (Mettes et al., 2019) for unified classification and Hyperspherical Variational Autoencoders (Davidson et al., 2018) for stable latent modeling. By ensuring transport remains geometrically consistent with the manifold, our approach enables high-capacity associative memory (Schlegel et al., 2022) in any system where semantics are encoded in vector direction rather than magnitude.

## 2. Related Works

**Continuous Cognitive Representations**  Vector Symbolic Architectures (VSAs) have recently evolved from a focus on discrete symbolic systems to robust frameworks for modeling continuous cognitive processes. Spatial Semantic Pointers (SSPs), in particular, have emerged as a leading method for encoding continuous variables into high-dimensional distributed representations (Komer et al., 2019). It was later shown that leveraging hexagonal grid bases to mimic the firing rates of grid cells optimize the accuracy of these representations (Dumont & Eliasmith, 2020). SSPs have also been used to model complex dynamical systems, allowing trajectory prediction of chaotic attractors (Voelker et al., 2021). This representational power has been applied to biological navigation (Komer & Eliasmith, 2020) and SLAM (Dumont et al., 2023), as well as modeling compositional relations in hippocampal-entorhinal circuits (Kymn et al., 2024). In all of these cases, clean up (denoising), plays a

crucial role in allowing robust performance of the networks.

While this past work demonstrates the utility of SSPs, they predominantly rely on classical cleanup mechanisms (e.g., discrete grid lookup or convex optimization) to handle noise. These methods often struggle with scalability and fail at higher noise regimes.

**Generative denoising as Associative Memory**  To address the limitations of classical cleanup, we look to the intersection of associative memory and generative modeling. Hoover et al. (2023) and Pham et al. (2025) have demonstrated that iterative generative denoising is theoretically equivalent to energy minimization in continuous attractor networks. This suggests that modern generative models can serve as the "dynamic cleanup" mechanism that VSAs have historically lacked. Indeed, recent workshops have explicitly called for bridging the gap between cognitive architectures and generative models (Furlong & Eliasmith, 2023), identifying that VSAs provide the necessary compositional structure that black-box generative models lack.

**Geometric Constraints and Manifold Learning**  Applying off-the-shelf generative models to SSPs is ineffective due to geometric mismatch. As argued in the Geometric Deep Learning blueprint (Bronstein et al., 2021), imposing Euclidean priors on non-Euclidean data (grids, groups, geodesics) leads to poor sample efficiency and broken symmetries. This issue is well-documented in robotics, where standard diffusion models fail to generate valid orientation trajectories because they violate the topology of the rotation group $SO(3)$ (Braun et al., 2024). To resolve this, we draw upon Riemannian Flow Matching, introduced by Chen & Lipman (2024). By defining probability paths via geodesics rather than linear interpolants, this framework allows for exact likelihood training on complex manifolds. While this approach has been successfully applied in domains such as robot motion planning (Braun et al., 2024) and protein folding (Yim et al., 2023), it has not yet been exploited for high dimensional structured representations like those used in VSAs. Our work is the first to extend this geometric formulation to high-dimensional spaces and topologies with non-gaussian statistics, bridging the gap between associative memory and valid neurosymbolic cleanup.

## 3. Background

### 3.1. Vector Symbolic Architectures (VSAs)

VSAs are representational frameworks that encode structured information into high-dimensional distributed vectors to enable symbolic manipulation through algebraic operations. We focus on Holographic Reduced Representations (HRRs), a type of VSA that associates concepts through binding (circular convolution) (Plate, 1995). Concepts are

represented using random high-dimensional vectors, allowing for compositionality and noise robustness under superposition through the following operations.

- **Similarity:** Measured via the dot product $\langle \psi, \gamma \rangle$.

- **Bundling (superposition):** Element-wise addition $\delta = \psi + \gamma$. The result is similar to both inputs, allowing for the representation of sets.

- **Binding (composition):** Circular convolution $\delta = \psi \circledast \gamma$. The result is quasi-orthogonal to the inputs but preserves information in a compressed format (e.g., assigning roles like COLOR $\circledast$ RED).

- **Unbinding:** An approximate inverse $\psi^{-1}$ retrieves information (e.g., $\delta \circledast \psi^{-1} \approx \gamma + \text{noise}$).

HRRs enable representation of complex structures (e.g., **OBJ** = SHAPE$\circledast$SQUARE+COLOR$\circledast$RED). However, unbinding operations introduce noise terms proportional to the number of stored items, necessitating robust cleanup mechanisms.

### 3.2. Spatial Semantic Pointers (SSPs)

Spatial Semantic Pointers (SSPs) extend HRRs by encoding continuous coordinates $x \in \mathbb{R}^m$ into high-dimensional vectors $\phi(x) \in \mathbb{R}^d$ via a frequency-based encoding. As defined by Dumont & Eliasmith (2020), the representation is constructed in the Fourier domain as a vector of phasors:

$$\tilde{\phi}(x)_j = e^{i\langle \theta_j, x \rangle}$$

where the encoding matrix $\Theta = [\theta_1, \ldots, \theta_d]^T \in \mathbb{R}^{d \times m}$ is composed of row vectors $\theta_j$ derived from scaled hexagonal grid bases (with conjugate symmetry). This construction ensures that the resulting real-valued vectors lie on a Clifford Hypertorus embedded within the unit hypersphere $\mathbb{S}^{d-1}$.

The dot product between two SSPs induces a similarity kernel that measures spatial proximity (Figure 1)

Locally, the SSP manifold is flat, and binding corresponds to addition in the embedded spatial domain:

$$\phi(x_1) \circledast \phi(x_2) = \mathcal{F}^{-1}\{e^{i\Theta \ell x_1} \odot e^{i\Theta \ell x_2}\} = \phi(x_1 + x_2)$$

A key advantage of SSPs is their ability to bind with other VSA representations to form *cognitive maps*. For example, consider encoding a cat at $(x_1, y_1)$, a mouse at $(x_2, y_2)$, and cheese at $(x_3, y_3)$:

$$\mathbf{M} = \mathbf{CAT} \circledast \phi(x_1, y_1) + \mathbf{MOUSE} \circledast \phi(x_2, y_2)$$

$$+\mathbf{CHEESE} \circledast \phi(x_3, y_3)$$

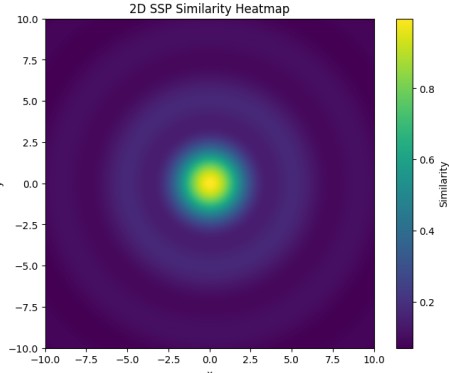

*Figure 1.* Spatial Semantic Pointer (SSP) representations ($d = 487$) of a 2D input shown by computing similarity of input to SSP embeddings of the domain.

This map can be queried for objects or their locations via approximate unbinding, e.g.,

$$\mathbf{M} \circledast \phi(x_3, y_3)^{-1} \approx \mathbf{CHEESE} + \epsilon, \qquad (1)$$

where $\epsilon$ denotes interference from the other terms. We demonstrate this kind of representation in a SLAM system in Section 5.4. This motivates the next section, which formally quantifies sources of noise in SSPs.

### 3.3. The Geometry of Noise

VSA operations and spiking neural implementations introduce noise that pushes vectors off the valid manifold, degrading the geometric properties required for retrieval.

**Bundling and Unbinding (cross-talk noise).** In a bundled memory trace/cognitive map, querying one item introduces interference from others (e.g. Equation 1). Relying on the quasi-orthogonality of the constituent vectors, this interference scales as a noise distribution $\epsilon \sim \mathcal{N}(0, \frac{n-1}{d}I_d)$. The per-component variance grows linearly with the number of stored items.

**Accumulating Phase Drift in Recurrence.** A critical source of noise arises from the recurrent connections used to implement path integration. When an SSP is updated over time, small inaccuracies in spike timing or synaptic transmission accumulate as phase errors. For a single Fourier component $\tilde{\phi}_j(x_t) = e^{i\langle \theta_j, x_t \rangle}$, the noisy update is:

$$\tilde{\phi}_j^{(t+1)} = \tilde{\phi}_j^{(t)} e^{i(\Delta \theta_{j,t} + \delta_{j,t})}, \quad \delta_{j,t} \sim \mathcal{N}(0, \sigma^2). \quad (2)$$

After $t$ steps, the cumulative perturbation is

$$\epsilon_{j,t} = \exp(i \sum_{\tau=1}^{t} \delta_{j,\tau})$$

Since the sum of errors is normally distributed $\sum \delta \sim \mathcal{N}(0, t\sigma^2)$, the resulting error term follows a Wrapped Normal distribution on the circle:

$$\epsilon_{j,t} \sim \text{WrappedNormal}(\mu = 0,\ \sigma^2 = t\sigma^2) \qquad (3)$$

**Neural Activity Noise (Spiking Variability).** Spiking neural networks exhibit another form of noise due to random spiking variability. To decode a represented value from a population, noise scales with dimensionality of the represented vector and inversely with the number of neurons in the population (i.e., $\epsilon \sim \mathcal{N}(0, \sigma^2 d/N_{tot})$).

### 3.4. The Geometric Gap: Euclidean Failure

Conditional Flow Matching (CFM) (Lipman et al., 2022) learns a deterministic time-dependent velocity field $v_\theta(\phi, t)$ that transports probability mass from a simple source distribution $p_0$ to a complex target distribution $p_1$. Inference is performed by integrating the learned Ordinary Differential Equation (ODE) from $t = 0$ to $t = 1$:

$$\frac{d\phi_t}{dt} = v_\theta(\phi_t, t), \quad \phi_0 \sim p_0. \qquad (4)$$

To train $v_\theta$, standard CFM defines a "conditional probability path" as a linear interpolation between a noise sample $\phi_0$ and a data sample $\phi_1$:

$$\phi_t = (1 - t)\phi_0 + t\phi_1. \qquad (5)$$

This path corresponds to a constant target velocity field $u_t$ that points directly from source to target:

$$u_t(\phi|\phi_0, \phi_1) = \frac{d}{dt}\phi_t = \phi_1 - \phi_0. \qquad (6)$$

The model is trained by minimizing the regression loss

$$\mathcal{L}_{CFM} = \mathbb{E}_{t,p_0,p_1}[\|v_\theta(\phi_t, t) - u_t\|^2]. \qquad (7)$$

While the linear interpolant $\phi_t$ is efficient for flat Euclidean spaces, it is geometrically invalid for data constrained to a hypersphere $\mathbb{S}^{d-1}$. A straight line connecting two points on a sphere is a chord that passes through the interior.

Consequently, for any intermediate time $t \in (0, 1)$, the vector magnitude collapses ($\|\phi_t\| < 1$). For high-dimensional SSPs, this traversal through the origin destroys the normalized phase structure ($\tilde{\phi}_j = e^{i\theta_j x}$) required to encode spatial semantics. To address this, we follow the approach taken by Chen & Lipman (2024) and propose Geodesic Flow Matching, which redefines the transport dynamics using Riemannian maps to ensure manifold consistency.

## 4. Methodology

We treat cleanup as a generative transport problem. We model the prior distribution $p_0$ as isotropic Gaussian noise projected onto the unit hypersphere: $\phi_0 = z/\|z\|$, where $z \sim \mathcal{N}(0, I_d)$. This approximates maximally corrupted SSPs (under the noise models discussed in Section 3.3).

The clean data (target distribution $p_1$) consists of valid Hexagonal SSP encodings $\phi(x)$ (as defined in Sec 3.2). Samples are drawn using Sobol quasi-random sequences to ensure uniform coverage of the spatial domain without grid artifacts.

### 4.1. Baselines

**Grid Lookup** discretizes the domain $\mathcal{D}$ into a finite grid $\mathcal{G}$ and computes the similarity $s(x) = \langle \phi(x), \tilde{\phi} \rangle$ against the corrupted input $\tilde{\phi}$. The method selects the candidate $x_{\text{init}} = \arg\max_{x \in \mathcal{G}} s(x)$, guaranteeing that the result lies on the valid manifold but limiting scalability due to the exponential cost of grid resolution.

**Optimization-Based Cleanup** refines the coarse grid estimate $x_0 = x_{\text{init}}$ through continuous optimization to maximize $J(x) = \langle \phi(x), \tilde{\phi} \rangle$. We employ the L-BFGS-B algorithm (Liu & Nocedal, 1989), which approximates the inverse Hessian $H_k^{-1}$ using a limited history of curvature pairs. The update step is given by $x_{k+1} = \text{Proj}_{\mathcal{D}}(x_k + \alpha_k H_k^{-1} \nabla_x J(x_k))$, where $\text{Proj}_{\mathcal{D}}$ ensures the state remains within valid bounds. This quasi-Newton approach provides near–second-order convergence with linear memory cost, though it relies on the initialization $x_{\text{init}}$ to avoid local maxima on the non-convex SSP manifold.

**Direct Forward Regression** is implemented by a network $f_\theta$ trained to map $\phi_0$ to $\phi_1$ directly. We optimize cosine similarity (Eq. 8) to focus on directional alignment.

$$\mathcal{L}_{FF} = \mathbb{E}_{\phi_0, \phi_1}\left[1 - \frac{f_\theta(\tilde{\phi})^\top \phi_{\text{clean}}}{|f_\theta(\tilde{\phi})|, |\phi_{\text{clean}}|}\right] \qquad (8)$$

As a global mapping, this approach lacks iterative refinement. We demonstrate that without the guidance of a flow, the model collapses to predicting "averages" of the target distribution (a bundle over the domain) rather than a specific clean instance.

**Euclidean Flow Matching (I-CFM):** is a form of Independent Conditional Flow Matching (I-CFM) that defines the probability path as a linear interpolation between noise and data: $\phi_t = (1 - t)\phi_0 + t\phi_1$. The target velocity field is the time-derivative of the path: $u_t(\phi|\phi_0, \phi_1) = \phi_1 - \phi_0$.

This chordal path "cuts through" the interior of the hyper-

sphere, causing vector magnitude to collapse and destroying the phase information required for training.

## 4.2. Geodesic Flow Matching

In our proposed method, we constrain the flow to the unit hypersphere $\mathcal{M} = \mathbb{S}^{d-1}$. Transport is defined using Riemannian maps that ensure all intermediate states $\phi_t$ remain on the manifold.

Instead of a straight line, the probability path follows the great-circle arc (geodesic) between source and target:

$$\phi_t = \text{Exp}_{\phi_0}(t \cdot \text{Log}_{\phi_0}(\phi_1)) \qquad (9)$$

$\text{Log}_{\phi_0}$ (Logarithmic Map) projects the target $\phi_1$ into the tangent space at $\phi_0$ ($T_{\phi_0}\mathbb{S}^{d-1}$), defining the initial velocity vector. $\text{Exp}_{\phi_0}$ (Exponential Map): Maps a tangent vector back onto the manifold, advancing the state along the geodesic.

A neural network is trained to regress a vector field $v_\theta(\phi_t, t)$ by predicting instantaneous tangent velocities ($\dot{\phi}_t$) along this arc:

$$u_t = \frac{d}{dt}\text{Exp}_{\phi_0}(t\,v)\Big|_{v=\text{Log}_{\phi_0}(\phi_1)}$$

This is the parallel-transported velocity along the geodesic, maintaining constant speed $\|v\|$ along the arc, ensuring predicted updates are always tangent to the hypersphere and preserving the unit-norm constraint by design. We train using cosine flow loss:

$$\mathcal{L}_{\cos} = \mathbb{E}_{t,\phi_0,\phi_1}\left[1 - \frac{v_\theta(\phi_t, t)^\top \dot{\phi}_t}{|v_\theta(\phi_t, t)||\dot{\phi}_t|}\right] \qquad (10)$$

Training and inference procedures are detailed in Algorithms 1 and 2.

---

**Algorithm 1** Training: Geodesic Flow Matching on $\mathbb{S}^{d-1}$

---

1: Sample $t \sim \mathcal{U}[0, 1]$ and $\phi_0, \phi_1 \sim \pi(\phi_0, \phi_1)$
2: $v = \text{Log}_{\phi_0}(\phi_1)$        ▷ Tangent direction at $\phi_0$
3: $\phi_t = \text{Exp}_{\phi_0}(t\,v)$        ▷ Geodesic interpolant
4: $u_{\text{true}} = \dfrac{d}{dt}\text{Exp}_{\phi_0}(t\,v)$    ▷ Target velocity at $\phi_t$
5: **return** $(\phi_t, t, u_{\text{true}})$

---

**Algorithm 2** Inference: Geodesic Sampling

---

**Require:** Trained field $v_\theta(\phi, t)$, step count $K$, step size $\Delta t = 1/K$
1: Initialize $\phi_0 \sim p_0$     ▷ Sample from uniform noise on $\mathbb{S}^{d-1}$
2: **for** $k = 0$ to $K - 1$ **do**
3:     $v_k = v_\theta(\phi_k, t_k)$
4:     $\phi_{k+1} = \text{Exp}_{\phi_k}(\Delta t\, v_k)$     ▷ Geodesic step
5:     $\phi_{k+1} \leftarrow \phi_{k+1}/\|\phi_{k+1}\|$    ▷ Numerical stability correction
6: **end for**
7: **return** $\phi_K$     ▷ Cleaned SSP sample

---

# 5. Experiments

## 5.1. Setup and Baselines

We compare our method against Euclidean Flow Matching (CFM), Feedforward Regression, a $64 \times 64$ Grid Lookup, and L-BFGS-B Optimization with a $4 \times 4$ grid lookup initialization. We sweep dimensionality $d \in \{55, \ldots, 1015\}$ to test scalability. Given pairings consisting of a clean SSP and hyperspherical noise samples, we generate intermediate noised samples $\phi_t$ using the Log/Exp maps introduced in Section 4.2. We parameterize $v_\theta$ using a Residual MLP with sinusoidal time embeddings.

The network $v_\theta$ is a 3-block Residual MLP. Each `ResBlock` consists of two linear layers with GELU activation, dropout ($p = 0.1$), and a residual connection normalised with LayerNorm. The hidden widths follow a bottleneck schedule: $2d \to d$, $4d \to d$, $2d \to d$, where $d$ is the SSP dimensionality. Time is encoded via a sinusoidal positional embedding of dimension 32 and concatenated to the input.

The Log map clamps $\langle p, q \rangle$ to $[-1, 1]$ before arccos and uses an $\epsilon = 10^{-8}$ norm floor on the perpendicular component, handling antipodal and coincident points respectively. The Exp map applies the same floor on $\|v\|$.

For each method, intermediate training samples $\phi_t$ are generated according to that method's own transport geometry: linear interpolants for Euclidean CFM (which are deliberately *not* re-projected onto $\mathbb{S}^{d-1}$, as this interior traversal is the failure mode under study) and geodesic interpolants (Eq. 9) for GFM, which remain on the manifold by construction.

## 5.2. Denoising Benchmarks

The quantitative benchmark shown in Figure 2 examines the various cleanup methods' ability to recover a ground truth SSP ($\phi_1$) under noise corruption by generating intermediate samples $\phi_t$ using Equation 9 by measuring the

cosine similarity between the output and ground truth. Results reveal that Euclidean Flow Matching degrades significantly as noise increases due to the linear interpolant "cutting through" the hypersphere.

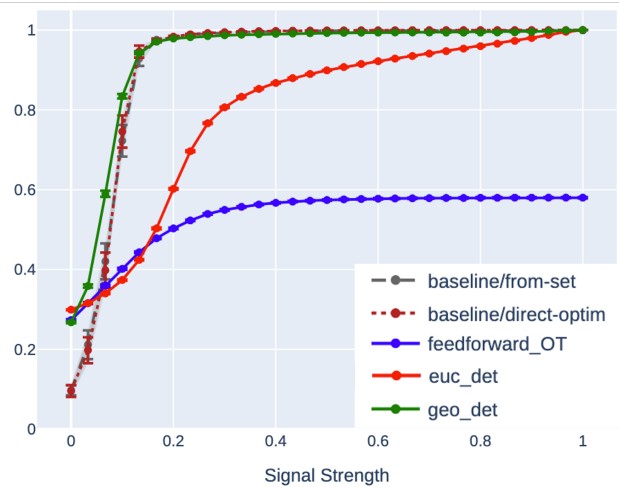

*Figure 2.* **Absolute Performance vs. Noise** ($d = 1015$). Note the degradation of Euclidean methods at high noise levels.

This is supported by the performance gap between Geodesic and Euclidean flows across dimensions (Figure 3). The benefit of Geodesic constraints increases sharply from low dimensions ($d \approx 50$) to moderate dimensions ($d \approx 200$) before stabilizing at a positive plateau ($d > 500$). Overall this graph suggests our method improves performance by $\sim 10\%$ versus the Euclidean baseline.

**Comparison to non-neural baselines:** At high signal strengths, non-neural baselines (Grid/Optim) perform near-perfectly (like GFM) due to the high separability of targets. However, in high-noise regimes (Signal Strength $< 0.15$), GFM significantly outperforms them. While discriminative baselines fail ("snapping" to random prototypes) once the signal drops below the noise floor, GFM uses the learned vector field as a geometric prior to continuously transport the noisy state toward the correct region within the manifold.

### 5.3. Qualitative Analysis: Manifold Structure

To better characterize the errors of past methods, we visually inspect the reconstruction quality by decoding the cleaned vectors back into the spatial domain using the SSP similarity kernel. Figure 4 presents a comparison of the inference outputs for the neural methods.

- **Geodesic Flow (Figure 4a):** The geodesic model reconstructs the target with high fidelity, producing a sharp, unimodal peak centered precisely at the correct spatial coordinate. This confirms that constraining

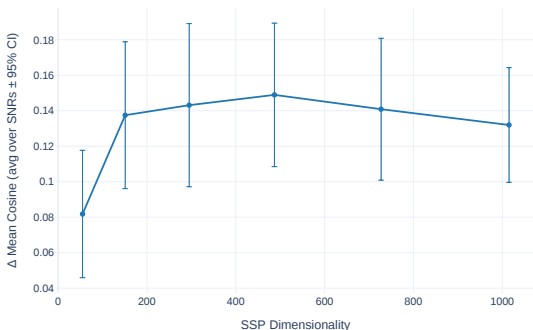

*Figure 3.* **Scalability Analysis.** Performance benefits of Geodesic interpolants relative to Euclidean ($\Delta$ mean cosine $\pm$ 95% CI) vs dimensionality. The advantage of the geodesic constraint is consistent and significant across the high-dimensional regime.

transport to the manifold preserves the phase relationships required for accurate localization.

- **Euclidean Flow (Figure 4b):** While the Euclidean model produces a vector that resembles a valid SSP (high similarity in a localized region), the peak is spatially displaced from the ground truth. We attribute this "drift" to the linear interpolant traversing the interior of the hypersphere. As the magnitude collapses during transport, the delicate phase information encoding the specific coordinates is corrupted, resulting in a valid-looking but semantically incorrect state.

- **Feedforward Regression (Figure 4c):** The direct feedforward network fails to commit to a single sharp location. Instead, it outputs a diffuse representation that resembles a superposition (bundle) of SSPs covering the region surrounding the target. Without the iterative guidance of a flow field to break symmetry, the regression objective collapses to the average of the posterior distribution, blurring the spatial estimate.

### 5.4. Application: Spiking Neural SLAM

Results from the previous section demonstrate superior performance of our GFM method, particularly in high noise regimes. To demonstrate the utility of this improvement, we now consider a continuous neurosymbolic reasoning benchmark: Semantic SLAM (Dumont et al., 2023), a system that performs simultaneous localization and mapping using VSA operations throughout. Unlike discrete neurosymbolic benchmarks that rely on prototype dictionaries, this task requires continuous VSA operations that are directly degraded by the noise sources in Section 3.3: velocity updates via binding ($\phi(x_{t+1}) = \phi(x_t) \circledast \phi(\Delta x)$), allocentric map construction via bundling ($\mathbf{M} = \sum_i \mathbf{f}_i \circledast \phi(x_i)$), and spatial queries via unbinding ($\mathbf{M} \circledast \phi(x)^{-1} \approx \mathbf{f} + \epsilon$). Without

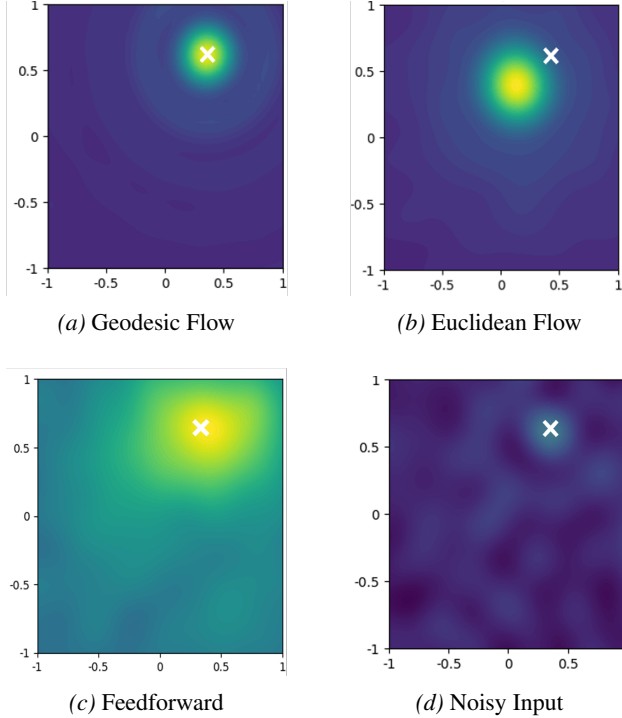

(a) Geodesic Flow        (b) Euclidean Flow

(c) Feedforward        (d) Noisy Input

*Figure 4.* **Qualitative Inference Comparison.** The Geodesic model (a) recovers a sharp peak at the target location. The Euclidean model (b) drifts, reconstructing a valid but misplaced SSP. The Feedforward model (c) produces a diffuse "bundle" covering the general region. (d) Shows the corrupted input state presented to each network before cleanup.

manifold-aware cleanup, accumulated noise collapses these symbolic operations.

Simultaneous Localization and Mapping (SLAM) requires an agent to track its position while building a map of landmarks. We adopt a biologically plausible architecture where position is tracked via Path Integration (PI) using a recurrent population of spiking neurons.

In spiking implementations, the PI accumulates phase errors over time due to neural variability and approximation errors as discussed in Section 3.3. Without correction, this drift causes the estimated position vector to leave the valid SSP manifold, leading to catastrophic failure in both localization and map construction.

We deploy the GFM cleanup model as an online stabilizer that intercepts the drifting PI output before it is used for landmark binding. By projecting the noisy state back onto the manifold, the cleanup ensures that high-fidelity vectors are written to the associative memory map, enabling successful loop closure even when the integrator's internal state degrades. We benchmark this against a Grid Baseline ($64 \times 64$ resolution), the closest matching baseline in terms of performance in our results section.

**Experimental Setup:** The agent navigates a 2D environment with 50 randomly distributed landmarks for 60 seconds. We simulate resource-constraint in the environments by varying the number of neurons in the PI population ($N \in [1000, 3000]$). Lower neuron counts serve as a proxy for lower signal-to-noise ratios, resulting in faster drift accumulation.

The three noise sources defined in Section 3.3 are all present in this experiment. *Phase drift in recurrence* accumulates through the velocity-controlled oscillators of the PI over 60 seconds of navigation. *Spiking variability* is introduced by the leaky integrate-and-fire (LIF) neurons in the PI population. *Cross-talk* arises during loop closure when querying the memory map ($\mathbf{M} \circledast \phi(x)^{-1}$) introduces interference from the bundled landmark vectors. The SLAM experiment therefore serves as the out-of-distribution evaluation for all three noise regimes simultaneously. All components (the SNN path integrator, cleanup model, and grid baseline) were simulated concurrently on a standard CPU; there is no hardware divide between the spiking and continuous components.

**Dynamic Stability:** Although the Grid baseline achieves high accuracy in static benchmarks (above), it fails in the dynamic SLAM task (Figure 5c). Because the Grid method "snaps" inputs to the nearest discrete prototype, it introduces small discontinuous jumps in the state estimate. These discontinuities disrupt the velocity integration dynamics of the recurrent network, causing the trajectory to diverge. In contrast, Geodesic Flow provides a continuous correction field that "pulls" the drifting state back to the manifold without introducing state discontinuities, allowing for stable long-term integration.

**Quantitative Results:** In a resource-constrained regime ($N = 1500$); see Figure 5d, Geodesic cleanup reduces the average path error from 0.249m (Grid) to 0.076m, a 72% reduction. As shown in Table 1, a 1,500-neuron system augmented with Geodesic cleanup achieves a tracking accuracy (0.076m) comparable to a baseline system using 2,500 neurons (0.083m). This indicates that manifold-aware cleanup offers a 40% reduction in the spiking neural resources required for stable path integration.

# 6. Conclusion

We demonstrated that the intrinsic geometry of high-dimensional representations is critical for robust cleanup. We showed that Euclidean interpolants fail to preserve phase information in Spatial Semantic Pointers (SSPs), whereas Geodesic Flow Matching (GFM) robustly restores representations by restricting transport to the Riemannian manifold.

This work shifts the paradigm of associative memory from a

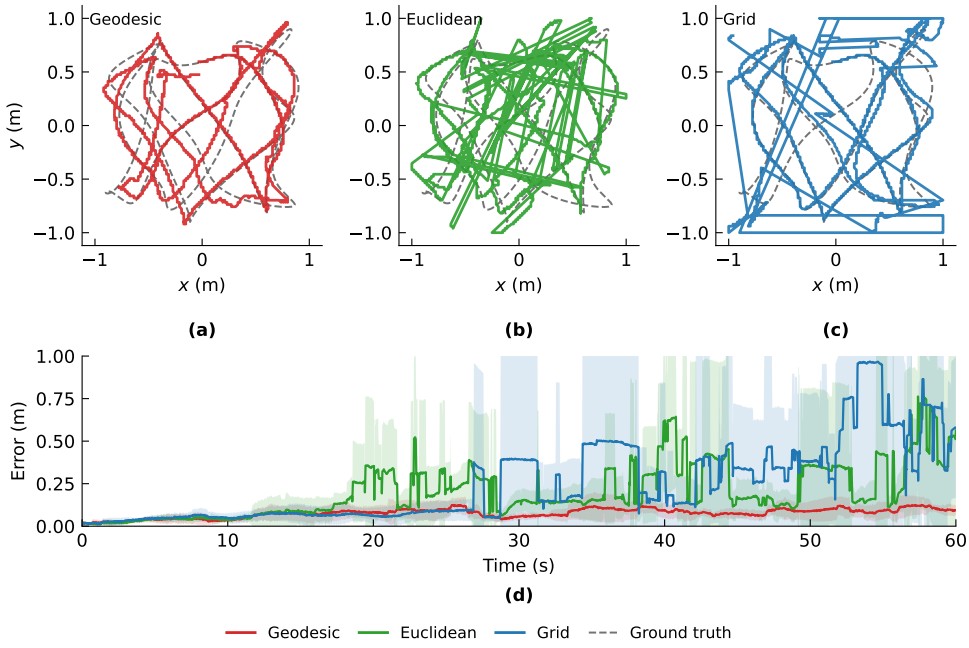

*Figure 5.* **SLAM Performance.** Trajectory reconstruction for a resource-constrained Path Integrator ($N = 1500$ neurons). **(a)** Geodesic Flow acts as a continuous attractor, maintaining a stable lock on the ground-truth trajectory. **(b)** Euclidean Flow introduces phase-destroying corrections that cause the estimated trajectory to diverge. **(c)** The grid baseline (Dumont et al., 2023) introduces discontinuous state jumps, also resulting in divergence. **(d)** Average aggregate error over 5 trajectories; Geodesic cleanup maintains low error throughout while both baselines accumulate drift.

*Table 1.* Comparison of Trajectory RMSE across Path Integrator Sizes. Bold indicates best performance.

| PI Neurons | Method | RMSE (m) |
|---|---|---|
| 1000 | Grid | $0.586 \pm 0.121$ |
| | Euclidean | $0.449 \pm 0.068$ |
| | Geodesic | $\mathbf{0.162 \pm 0.055}$ |
| 1500 | Grid | $0.249 \pm 0.239$ |
| | Euclidean | $0.204 \pm 0.103$ |
| | Geodesic | $\mathbf{0.076 \pm 0.026}$ |
| 2000 | Grid | $\mathbf{0.061 \pm 0.011}$ |
| | Euclidean | $0.599 \pm 0.153$ |
| | Geodesic | $0.067 \pm 0.014$ |
| 2500 | Grid | $0.083 \pm 0.017$ |
| | Euclidean | $0.190 \pm 0.089$ |
| | Geodesic | $\mathbf{0.078 \pm 0.009}$ |
| 3000 | Grid | $0.070 \pm 0.010$ |
| | Euclidean | $0.172 \pm 0.125$ |
| | Geodesic | $\mathbf{0.064 \pm 0.011}$ |

discrete search in the embedded domain to a continuous generative transport process directly in the embedding space.

Integration into a closed-loop Spiking Semantic SLAM system demonstrated that GFM stabilizes path integration against phase drift. This enabled a resource-constrained population (1,500 neurons) to match the performance of a much larger baseline (2,500 neurons), proving that manifold-aware cleanup enables efficient neuromorphic computing.

## Limitations and Future Work

This work opens several directions for future research. First, deployment on physical neuromorphic hardware requires converting the Residual MLP velocity field into a fully spiking network; established libraries such as snnTorch provide a straightforward pathway for this conversion. Second, the current architecture was chosen to validate the geometric transport formulation rather than to minimize compute; network optimization (layer type, sparsity, etc.) for resource-constrained environments remains future work. Third, the present formulation is specific to hyperspherical topology, which is a fundamental requirement of HRRs and SSPs. Extending geodesic generative transport to other VSA families with different underlying metrics (e.g., the Boolean hypercube $\{-1, +1\}^d$, or complex-valued Fourier-HRRs on $(\mathbb{S}^1)^{d/2}$) is a compelling direction that the general flow matching framework should support given a valid metric and transport trajectories.

## Acknowledgements

The authors would like to thank Nicole Dumont and P. Michael Furlong for discussions that helped improve this paper. This work was supported by CFI (52479-10006) and

OIT (35768) infrastructure funding as well as the Canada Research Chairs program, NSERC Discovery grant 261453, and AFOSR grant FA9550-17-1-0644.

## Impact Statement

This paper presents advances in machine learning for neurosymbolic AI. We foresee no direct negative societal impacts from this work. More broadly, advances that improve the robustness of continuous reasoning systems may eventually contribute to autonomous systems; the responsible deployment of such systems remains an important consideration for the field.

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
