# OpenReview forum: "Geodesic Flow Matching for Denoising High-Dimensional Structured Representations"
_ICML.cc/2026/Conference — ICML 2026 regular_

### Official Review · Reviewer_DU83 · 2026-03-09

**Soundness:** 2
**Presentation:** 2
**Significance:** 1
**Originality:** 2
**Overall Recommendation:** 2
**Confidence:** 3

**Summary:**

In this paper, the authors address the challenge of denoising high-dimensional neural representations, specifically Spatial Semantic Pointers (SSPs),  which lie on the unit hypersphere. The authors identify that conventional Euclidean Conditional Flow Matching (CFM) are geometrically invalid for this manifold, as linear paths leave the hypersphere and destroy the semantic structure. So they propose the Geodesic Flow Matching (GFM) , which is an adapted version of flow matching  to constrain the generative process to the hypersphere using logarithmic and exponential maps. The algorithm is evaluated in a denoising task for  spiking neural SLAM, and compared with Euclidean Flow Matching (CFM), Feedforward Regression, Grid Lookup and L-BFGS-B Optimization. The experiment results showed GFM achieved a 72% reduction in path error and a 40% increase in neural efficiency compared to competitive baselines.

**Compliance With Llm Reviewing Policy:**

Affirmed.

**Key Questions For Authors:**

Q1: The paper does not specify the architecture used for this key component. Could you provide details on the network design (e.g., number of layers, hidden dimensions, activation functions, whether you used any geometric inductive biases or invariance properties)?

Q2: For Line 57 and 58, By constraining transport to the Riemannian manifold using logarithmic and exponential maps, we ensure the cleanup process remains geometrically consistent. Regarding the numerical implementation of the logarithmic and exponential maps on the high-dimensional sphere (Lines 57-58): The exponential map, requires normalizing the tangent vector by its norm, This involves a division by $\|v\|$, which can be numerically unstable when the norm approaches zero. Conversely, the logarithmic map, $\log_{\phi_0}(\phi_1)$,  can be problematic when the value is near-zero. Could you elaborate on the numerical strategies or stabilizations (e.g., epsilon tolerances, handling of antipodal points) you implemented to ensure stable training and inference in these edge cases?

**Limitations:**

Yes

**Strengths And Weaknesses:**

Strengths:
The authors tackled a very practical problem that classical flow matching would fail, they extend RFM on SSP, and provide the algorithm to address the obstacle. They tested the algorithm in the real application as well.

Weaknesses:
The contribution of this paper is incremental, which is a direct adaptation of the existing Riemannian Flow Matching (RFM) framework (Chen & Lipman, 2023) to the specific manifold. While the extension to high-dimensional, structured representations is non-trivial, the fundamental mathematical machinery (log/exp maps on the sphere) is well-established. The novelty is therefore more in the application domain than in the development of a new generative modeling technique.
The neural network architecture is not described in detail. It is unclear whether the high-dimensional nature of the problem required specific architectural considerations (e.g., graph neural networks, transformers, or specific invariant layers) or if a standard MLP was sufficient. I would suggest the authors add this.

---

### Official Review · Reviewer_SEJ4 · 2026-03-11

**Soundness:** 2
**Presentation:** 2
**Significance:** 3
**Originality:** 3
**Overall Recommendation:** 3
**Confidence:** 2

**Summary:**

The submission proposes a cleanup method for Spatial Semantic Pointers (SSPs) using geodesic flow matching rather than standard Euclidean flow matching. The core claim is that linear interpolants used in Euclidean flow matching pass through the interior of the hypersphere and disrupt the phase relations needed for accurate SSP decoding. To avoid that, the method uses logarithmic and exponential maps to define transport on a manifold during denoising.
The paper evaluates the method in two settings. First, it reports denoising benchmarks across noise levels and ambient dimensions. Second, it inserts the cleanup model into a spiking-neural SLAM system and reports lower trajectory error under resource-constrained conditions. The reported headline numbers are a 72% reduction in path RMSE relative to a grid baseline at one operating point, and a claimed 40% reduction in neural resource requirements.
My overall view is that the paper contains a useful direction, but the empirical evaluation has several weaknesses, among other not testing for cross-talk, phase drift in recurrence, and spiking variability, and claiming a 40% reduction in cost is not fully supported.

**Compliance With Llm Reviewing Policy:**

Affirmed.

**Key Questions For Authors:**

1. How should the reader reconcile the target velocity definition in Section 4.2 with Algorithm 1?
2. What is the inference cost of cleanup, including step count and latency?

**Limitations:**

*	Runtime and compute cost are not reported, but efficiency gains are still claimed.
•	The method depends on a clear geometric structure, so it may be harder to apply to other types of representations unless similar mappings exist.

**Strengths And Weaknesses:**

Strengths:
•	The paper connects SSP cleanup with flow-based generative modeling in a way that is easy to follow.
•	The intuition for why Euclidean straight-line interpolation can be problematic for normalized, phase-based representations is clear.
•	The paper includes an application-level experiment in spiking SLAM .

•	The qualitative reconstructions are informative and help explain the failure modes of the baselines.

Weaknesses:
* Inconsistencies in the target velocity:
Section 4.2 states that the network predicts $u_t = \Log_{\phi_t}(\phi_1).$

However, Algorithm 1 defines the target velocity as $u_{\mathrm{true}} = \frac{d}{dt}\,\Exp_{\phi_0}(t v),$
which is described as the target velocity at \(\phi_t\).

Please explain weather \(u_t\) and \(u_{\mathrm{true}}\) are intended to represent the same quantity.

In particular, the use of \(\Exp\) here is not consistent with the earlier definition based on \(\Log\). If the model is meant to predict the tangent vector from \(\phi_t\) toward \(\phi_1\), then the target should be written in a way that matches that interpretation.

* The denoising experiments appear to generate intermediate samples using the same geodesic path family used during training. That makes the evaluation favorable to the method by construction.
Section 3.3 lists concrete SSP noise sources: cross-talk, phase drift in recurrence, and spiking variability, but this is not part of the evaluation. A stronger evaluation would include explicit cross-talk, phase drifts, etc., at least an explanation of how this is part of the data

* Figure 6 confuses me. Text refers to 6b, but there is no b-part in figure 6.
In the caption it says that the left is the geoside flow, but the text on the left part of the figure says "Euclidean flow." How can that be correct?

* The efficiency claim compares a smaller path integrator plus cleanup against a larger baseline path integrator. But the cleanup network itself has a computational cost, and that cost is not included in the efficiency accounting. Since the paper motivates this system partly in  efficiency,  the comparison should include: cleanup network compute, inference step count, latency, and maybe even energy usage.

---

> ### Author Rebuttal · Authors · 2026-03-31
>
> Thank you for your rigorous review. We appreciate the opportunity to clarify our notation, baselines, and evaluation design. Below we address your concerns.
>
> 1. Inconsistencies in the target velocity notation
>
> Thank you for catching this mathematical imprecision in the text. You are correct that the two quantities differ by a scalar factor relating to the remaining integration time. $Log_{\phi_t}(\phi_1)$ yields the vector required to traverse the remaining distance in $t=1$ time, whereas the constant-speed target velocity along the geodesic is $u_{true} = \frac{d}{dt} Exp_{\phi_0}(t v)$.
> Our implementation (and Algorithm 1) correctly computes the analytical time-derivative of the exponential map (the parallel-transported velocity), as this maintains the constant speed $||v||$ along the arc. We have corrected the notation in Section 4.2 to strictly match Algorithm 1 and our codebase, using the proper derivative definition.
>
> 2. Denoising evaluation bias
>
> We apologize for the confusion caused by the phrasing in Section 5.1; the evaluation was not biased toward our method. In the static denoising benchmark, the Euclidean CFM model was trained and evaluated using samples generated via linear interpolation (its own assumed noise model). GFM was trained and evaluated using geodesic interpolation. Neither method was forced to denoise paths generated by the opposing model's geometry. We will rewrite Section 5.1 for the camera-ready version to explicitly clarify that intermediate samples for each method were generated according to their respective transport paths.
>
> 3. Explicit noise sources in the evaluation
>
> We agree that it was unclear which noise sources were evaluated. In the updated draft, we will be clear that the dynamic SLAM experiment (Section 5.4) is the explicit, out-of-distribution evaluation of the noise sources defined in Section 3.3. We will expand Section 5.4 to explicitly map these theoretical noise sources to the system dynamics:
> - Phase drift in recurrence: The recurrent velocity-controlled oscillators continuously inject accumulating phase errors during spatial tracking.
> - Spiking variability: Simulated LIF neurons introduce activity noise into the position estimate.
> - Cross-talk: Querying the memory map ($\sum_i \phi(x_i) \otimes B_i$) during loop closure introduces interference from bundled representations.
>
> GFM acts as an episodic stabilizer against these forces. By projecting the drifted state back to the valid manifold without discrete jumps seen in grid lookups, it creates a virtuous cycle: a stable path integrator builds an accurate map, which in turn provides precise spatial targets for future loop closures.
>
> 4. Figure 6 formatting and typos
>
> Thank you for pointing these out. The "Euclidean Flow" label on the left panel is indeed a typo and will be corrected to "Geodesic Flow." We will also update the text references to match the figure panels (e.g., using "Figure 6, Middle" instead of "6b"). Finally, results from the Euclidean flow model will also be included as further comparison with the geodesic approach.
>
> 5. Efficiency accounting and computational overhead
>
> To clarify our comparison: the efficiency claim evaluates a smaller Path Integrator + our proposed GFM cleanup against a larger Path Integrator + the baseline Grid cleanup. Because the baseline system also requires a cleanup mechanism to function, it is not computationally free.
> To directly answer your question regarding the inference cost: the GFM cleanup requires 5 function evaluations (steps) of the Residual MLP. We completely agree with the reviewer that a full system-level accounting of computational cost (latency, memory, and step count) is important for validating efficiency claims. For the camera-ready version, we will include an explicit compute benchmark in the Appendix comparing the entire proposed pipeline (the 1,500-neuron PI + GFM) against the baseline pipeline (the 2,500-neuron PI + Grid Lookup).
> Ultimately, because a cleanup mechanism is inherently required during loop closure, replacing a large, continuously active recurrent PI population and an unscalable grid cleanup with a smaller PI population supported by an episodic, fixed-size MLP represents a fundamental gain for system scalability that our full-system benchmarks will explicitly quantify.
>
> 6. Dependence on a clear geometric structure
>
> We agree that this formulation relies on hyperspherical topology, but this is a fundamental mathematical requirement of Holographic Reduced Representations (HRRs). For dot-product similarity to act as a valid spatial kernel, the vectors must reside on the surface of $\mathbb{S}^{d-1}$. However, the underlying mechanics of flow matching are highly adaptable. Should other Vector Symbolic Architectures use different topologies, the generative transport approach extends naturally, provided a valid metric and appropriate flow trajectories can be defined for that space. We will note this adaptability as an avenue for future work.

---

> > ### Author Rebuttal · Reviewer_SEJ4 · 2026-03-31
> >
> > 1. I believe this is resolved.
> > 2. If this section is rewritten and made clearer, this is good for me.
> > 3. Resolved if this is added.
> > 4. Again, if the typo is solved, then great.
> > 5. Ok, this might have been unclear to me while first reading. I believe that a more explicit compute benchmark may make it clearer.
> > 6. This was probably unclear to me due to the lack of familiarity with HRRs. I trust this is correct.

---

### Official Review · Reviewer_o7QV · 2026-03-11

**Soundness:** 2
**Presentation:** 3
**Significance:** 2
**Originality:** 3
**Overall Recommendation:** 3
**Confidence:** 3

**Summary:**

The paper introduces a novel geometric approach to cleaning up noisy, continuous data in neurosymbolic AI. Because SSPs are mathematically constrained to the surface of a hypersphere ($\mathbb{S}^{d-1}$), standard generative denoising techniques like Euclidean Flow Matching fail. Euclidean methods draw a straight line between a noisy state and a clean state, which essentially cuts through the empty interior of the hypersphere, destroying the vector's magnitude and delicate spatial phase relationships.To solve this, the authors propose Geodesic Flow Matching (GFM). By using Riemannian transport dynamics (Logarithmic and Exponential maps), GFM forces the denoising process to travel along the curved surface of the manifold (great-circle arcs). The authors validate this on a Spiking Neural SLAM (Simultaneous Localization and Mapping) system, demonstrating that GFM acts as a highly effective continuous attractor.

**Compliance With Llm Reviewing Policy:**

Affirmed.

**Final Justification:**

I would like to remain my score.

**Key Questions For Authors:**

1. Is the "Euclidean Flow" label in the left panel of Figure 6 a typo, given that the figure is meant to compare the Geodesic Flow model with the Grid Baseline? If it is a typo, could you also elaborate on the actual SLAM performance differences between the Euclidean and Geodesic flow models?

2. Section 5.1 states that the Grid Lookup baseline uses a dense $64 \times 64$ grid. However, the L-BFGS-B Optimization baseline is initialized using only a $4 \times 4$ grid lookup. Given that the SSP manifold is highly non-convex, a 16-point initialization seems almost guaranteed to trap the optimizer in a local maximum. Can the authors validate the effectiveness of this initialization?

3. How exactly is the Geodesic Flow Matching model integrated with the Spiking Neural Network in the SLAM experiment? The application section highlights a 40% reduction in neural resource requirements (1,500 vs 2,500 neurons) for the path integrator. However, if the continuous, real-valued flow matching model has to run on a standard GPU/CPU alongside the SNN to perform the cleanup, the overall system energy and computational costs might actually increase.

**Limitations:**

The authors have not included a dedicated discussion of the limitations. There are three potential limitations that are not discussed: the inference latency introduced by solving an ODE and computing Riemannian maps, the practical limitations of integrating a continuous, real-valued Residual MLP with a discrete Spiking Neural Network, and whether the current mathematical formulation is strictly limited to hyperspherical geometries or what adaptations would be required for other non-Euclidean spaces used in neurosymbolic AI.

**Strengths And Weaknesses:**

**Strengths**

The paper does an excellent job of isolating exactly why standard generative models fail on this type of data. The explanation of Euclidean linear interpolants "cutting through" the hypersphere and destroying phase structure is intuitive and mathematically sound. Instead of stopping at mathematical proofs or toy datasets, the authors test the method on a real application: Spiking Neural SLAM.

**Weaknesses**

1. Limited Application Scope: The paper's methodology relies strictly on the hyperspherical geometry inherent to SSPs. While the proposed GFM approach works flawlessly within this domain, the study lacks an exploration of its adaptability. It remains unclear whether this framework can be seamlessly translated to the more complex, non-spherical topologies utilized in other neurosymbolic architectures.

2. Computational Overhead Not Fully Explored: Flow matching requires iteratively evaluating a neural network to solve an Ordinary Differential Equation (ODE) during inference. While the authors note that it is faster than traditional diffusion models, they do not thoroughly benchmark the real-time inference latency of computing Riemannian Exponential and Logarithmic maps at high dimensions ($d>1000$) compared to the non-neural baselines.

---

> ### Author Rebuttal · Authors · 2026-03-31
>
> We thank the reviewer for highlighting areas where our experimental setup required further clarification. We address these specific points below.
>
> 1. Application Scope and Adaptability
>
> As correctly noted, hyperspherical geometry is a fundamental mathematical requirement of HRRs and SSPs. However, regarding the framework's adaptability to other neurosymbolic architectures: the underlying mechanics of flow matching are highly general. As long as a valid metric and appropriate flow trajectories can be defined for a given topology, this geodesic generative transport approach would extend to those spaces. We agree that exploring this adaptability across different VSAs is a compelling direction for future research.
>
> 2. Computational Overhead and Riemannian Maps
>
> Regarding the overhead of the Riemannian maps at high dimensions (d>1000): the Exponential and Logarithmic maps on the hypersphere consist entirely of highly efficient, element-wise O(d) operations (vector norms, dot products, and basic trigonometry). This mathematical overhead is negligible compared to the matrix multiplications of the neural network's forward pass.
>
> While evaluating an ODE over a few steps is exponentially faster than the hundred of steps in SDE integration required by diffusion models, evaluating a neural network does introduce latency compared to a discrete grid lookup. We used a standard Residual MLP as a baseline proof-of-concept because it successfully solved the geometric transport problem. Optimizing the specific network architecture (layer types, capacity, sparsity) to minimize the compute overhead of such a network is an important separate design axis that we will pursue as future work.
>
> 3. Integration with the SNN and Efficiency Claims
>
> To clarify the experimental setup: there is no hardware divide in our evaluations. The entire simulation (including the SNN path integrator, the baseline grid cleanup, and our GFM model) was executed concurrently on a standard CPU. We will make that clearer in the final manuscript.
>
> Regarding the 40% reduction in neural resources, this refers specifically to the size of the recurrent spiking population needed for stable path integration. In our response to reviewer 1, we note that we will provide a more detailed breakdown of the SLAM model, from which it should be more evident that this comes from changing the size of a particular module in the model. As discussed in our "Dynamic Stability" and "Quantitative Results" sections, replacing this module with GFM provides a continuous correction field that avoids the discrete state jumps introduced by the grid baseline. This superior stabilization is what allows a 1,500 neuron system with GFM to achieve tracking accuracy (0.076m) comparable to a 2,500 neuron baseline system (0.083m). For actual deployment onto physical neuromorphic hardware, the continuous real-valued MLP can be converted into a fully spiking network using established libraries like snnTorch.
>
> 4. Figure 6 Typo and Euclidean Performance
>
> We appreciate this error being pointed out; the "Euclidean Flow" label on the left panel is a typo and should read "Geodesic Flow." We will correct this. Furthermore, we will add the Euclidean Flow's performance to the SLAM trajectory plots in the camera-ready version to further highlight the performance boost brought by the GFM.
>
> 5. L-BFGS-B Optimization Initialization
>
> We understand the reviewer’s intuition, and have run ablation experiments showing performance as a function of the initialization. Given those results, the coarse 4×4 initialization was chosen deliberately. The SSP similarity manifold is highly non-convex, behaving like a sinc function with rippling local maxima. If we initialize the continuous optimizer with a dense 64×64 grid, the initialization step effectively solves the task via brute force lookup, converting the optimizer back into the Grid baseline. We used a coarse initialization to evaluate the continuous optimizer's actual ability to traverse the manifold. As shown in Figure 3, the grid baseline (from-set) perfectly matched the optimization method with coarse initialization. Increases in initialization resolution showed no impact on evaluation performance. We are happy to include those ablations with denser initializations in the appendix if it would provide further clarity for readers.
>
> 6. Addition of a Limitations Section
>
> We agree that explicitly outlining the boundaries of this proof-of-concept is crucial. We will add a "Limitations and Future Work" section to the conclusion. This section will specifically discuss: (1) Conversion of the model for a fully deployed SNN method on neuromorphic hardware; (2) network architecture optimization for resource constrained environments; and (3) exploring the adaptation of this method to other VSAs with different underlying metrics.

---

> > ### Author Rebuttal · Reviewer_o7QV · 2026-04-02
> >
> > Thank you to the authors for the rebuttal. My primary concerns regarding the application scope and adaptability remain unresolved. While you state that 'the underlying mechanics of flow matching are highly general' and would extend to other spaces as long as a valid metric is defined, defining such a metric for complex, real-world topologies is often intractable. The claim that this framework seamlessly generalizes relies entirely on an assumption that currently lacks both rigorous theoretical justification and empirical validation. Therefore, I would like to remain my score.

---

### Official Review · Reviewer_EorQ · 2026-03-13

**Soundness:** 2
**Presentation:** 3
**Significance:** 2
**Originality:** 3
**Overall Recommendation:** 4
**Confidence:** 3

**Summary:**

The paper introduces Geodesic Flow Matching (GFM), a method that builds upon Spatial Semantic Pointers (SSPs), which are used for symbolic reasoning in neurosymbolic AI. The authors show that Euclidean Flow Matching fails for high-dimensional SSPs and therefore adapt Flow Matching to high-dimensional geodesics.

**Compliance With Llm Reviewing Policy:**

Affirmed.

**Final Justification:**

I increased my score since the rebuttal resolved most of the weaknesses.

**Key Questions For Authors:**

- Do the authors have an hypothesis why the performance improvement compared to Euclidean CFM peaks at a SSP dimensionality of roughly 500 and decreases for larger dimensionalities (Figure 4)?
- How does the method perform on symbolic reasoning tasks like analogical reasoning?

**Limitations:**

yes

**Strengths And Weaknesses:**

Strengths:
- The method is well-motivated since the linear interpolants corresponding to Euclidean Flow Matching do not preserve the geometry of hyperspherical embeddings.
- The paper includes informative experiments, such as the comparison across dimensionalities (Figure 4) and the qualitative decoding analysis (Figure 5) both help characterize where and why GFM outperforms baselines.
- The paper is well-written and well structured.

Weaknesses:
- The method is not evaluated on symbolic reasoning benchmarks despite the introduction highlighting symbolic reasoning. A comparison of GFM to related methods on symbolic reasoning (e.g. analogical reasoning (Plate, 1995) or spatial question answering) would significantly improve the paper.
- SLAM experiment scope is limited. Only one architecture variant (Residual MLP with sinusoidal embeddings) is tested. There is no comparison to diffusion-based cleanup or to prior SSP-specific methods from the spiking SLAM literature (e.g., Dumont et al., 2023).
- The ~10% denoising improvement is modest. Figure 4 shows a performance gain of roughly 0.05–0.18 over Euclidean CFM across dimensions. While statistically significant, this is small in absolute terms.
- Minor formatting issue: The running title at the top of each page is still from the template.
- Minor formatting issue: I recommend using \citet or \citeauthor, see e.g. line 17 right.

---

> ### Author Rebuttal · Authors · 2026-03-31
>
> We thank you for your feedback and careful review. Below, we address your concerns and outline improvements for the manuscript.
>
> 1. Symbolic reasoning benchmarks:
>
> We agree that neurosymbolic benchmarks are important, and apologize for it not being clear that we are using a neurosymbolic benchmark.  Specifically, we use Dumont et al.'s (2023) Semantic SLAM as a benchmark because it includes neurosymbolic representations (hence ‘semantic’) during processing. We have updated the manuscript to more clearly identify this connection to avoid misunderstanding.
>
> Unlike discrete VSA methods (e.g., Hopfield networks) that rely on unscalable prototype dictionaries, GFM learns the continuous manifold, enabling infinite spatial resolution. The SLAM architecture relies on continuous VSA operations including:
>
> Velocity Updates via binding: $\mathcal{F}^{-1}\{e^{iAx(t)} \odot iA\dot{x}(t)\}$.
>
> Allocentric Map Generation via binding position and egocentric features: $\hat{\phi}(x(t)) \otimes \phi(x_i - x(t))$.
>
> Map Queries via bundling/binding: $\sum_i \phi(x_i) \otimes B_i$.
>
> Here, $B_i$ are neurosymbolic feature vectors (objects in spatial memory). Without GFM's manifold-aware recovery, downstream symbolic operations collapse under accumulated noise. We have updated the manuscript to explicitly frame this as continuous neurosymbolic reasoning and relabeled Figure 6 to "Grid Baseline (Dumont et al., 2023)".
>
> 2. Limitations of SLAM experiment scope:
>
> Regarding the Residual MLP: Our core contribution is a geometric transport formulation. While altering layers and architectural modifications could yield task-specific gains, our focus is ensuring the transport path respects the continuous manifold's geometric constraints, which we expect to be useful regardless of architectural details (although this remains to be seen for a wide set of architectures).
>
> Regarding a diffusion baseline: Riemannian diffusion faces two bottlenecks here. First, solving SDEs over hundreds of steps violates real-time SLAM latency constraints. Second, scaling to high-dimensional tori ($d > 500$) requires implicit score matching. This relies on stochastic divergence estimation, introducing significant variance and instability (Lipman et al., 2023). Flow matching bypasses this via fast, deterministic ODEs. However, should the reviewer believe a direct comparison would add valuable context, we would be happy to run a diffusion baseline and include it in the final manuscript.
>
> 3. Impact of \~10\% improvement in  denoising:
>
> While we agree that a 10\% gain in cosine similarity may appear numerically modest, we believe we have shown that it can have significant functional implications. As shown in Figure 5b, the Euclidean flow generates a vector with high overall similarity,  but the loss of phase information causes it to recover the wrong spatial location. For the camera ready version, we will further emphasize that this effect is more pronounced for higher noise levels (i.e., recovered locations are further from ground truth due to ambiguity in the reconstruction process).
>
> In a recurrent system, recovering an incorrect spatial representation (even one that looks mathematically "decent" in terms of raw similarity to a represented location) leads to catastrophic drift. This is visually confirmed in the SLAM loop closure (Figure 6), where small deviations eventually collapse the self-location representation. GFM circumvents this phase destruction entirely. We will include the Euclidean CFM results in the SLAM evaluation for the camera-ready version to explicitly demonstrate this failure mode.
>
> 4. Minor formatting issues
> We thank the reviewer for pointing these out. Both the running title and citation formatting will be corrected in the camera-ready version
>
> 5. Hypothesis regarding the performance peak at $d \approx 500$
>
> To clarify, Figure 4 plots the average performance across all evaluated SNRs. The peak stems from how dimensionality interacts with the minimum SNR needed for signal recovery.
>
> Because of the distributed nature of SSPs, the SNR threshold for recovery decreases as dimensionality grows. As dimensionality increases up to $d \approx 500$, the geodesic model rapidly capitalizes on this dropping SNR threshold, achieving perfect reconstruction across a much wider range of noise levels, which maximizes the average performance gap over the Euclidean model. Beyond $d > 500$, the Euclidean model also begins to benefit from this distributed robustness, slightly improving its own average recovery rate. However, the geodesic method is already at maximum performance, so the difference in performance lessens. Nevertheless, because the Euclidean path fundamentally cuts through the ambient space, it always suffers from phase loss. Therefore, it never closes the gap completely, resulting in the stabilized performance difference seen at high dimensions.

---

> > ### Author Rebuttal · Reviewer_EorQ · 2026-04-04
> >
> > Thank you for the detailed rebuttal. The connection between semantic SLAM and symbolic reasoning was unclear to me. The rebuttal resolves this and the planned changes in the paper will likely improve this as well. An additional experiment that highlights the described bottlenecks of Riemannian diffusion would improve the work further. The arguments regarding (1) modest improvement in denoising vs. significant functional improvements and (2) performance across evaluated SNRs are convincing. Overall, I'm considering to increase my score.

---

### Decision · Program_Chairs · 2026-04-30

**Decision:**

Accept (regular)

**Comment:**

This paper extends Remannian flow matching to the manifold of Spatial Semantic Pointers, an embedding space for neurosymbolic reasoning, enabling generative models in SSP space.  Reviewers found the approach well-motivated and explanation of how regular FM fails for hyperspherical representations good. They also noted the innovative experiments and analysis.  Several concerns came up in the reviews which were largely addressed by the rebuttals including the value of the semantic SLAM experiments, the scale of the improvement, the computational overhead, and the mathematical formulation of the velocity field.  There were two unaddressed concerns: limited novelty and the limited scope.  In my opinion, the novelty is in the application and the scope is appropriate for the application.  It would have been good for the authors to directly respond to DU83 and while SEJ4 noted most of their concerns were addressed, they did not change their score to acceptance or list a final justification.